# Association between Ready-to-Eat Cereal Consumption and Nutrient Intake, Nutritional Adequacy, and Diet Quality among Infants, Toddlers, and Children in the National Health and Nutrition Examination Survey 2015–2016

**DOI:** 10.3390/nu11091989

**Published:** 2019-08-23

**Authors:** Jessica D. Smith, Yong Zhu, Vipra Vanage, Neha Jain, Norton Holschuh, Anne Hermetet Agler

**Affiliations:** 1Bell Institute of Health and Nutrition, General Mills, Inc., Minneapolis, MN 55427, USA; 2Global Knowledge Services, General Mills India Pvt. Ltd., Mumbai, Maharashtra 400076, India; 3Global Knowledge Solutions, General Mills, Inc., Minneapolis, MN 55427, USA

**Keywords:** ready-to-eat cereal, Healthy Eating Index 2015, children, dietary reference intakes, whole grains, nutrients of public health concern, food groups

## Abstract

Ready-to-eat (RTE) cereal is a popular food among children. However, there are no recent data on the associations between RTE cereal consumption and dietary outcomes in the U.S. Therefore, we sought to investigate how RTE cereal was associated with nutrient and food group intakes and overall dietary quality among children aged 0.5 to 17 years using the latest data from the National Health and Nutrition Examination Survey (NHANES 2015–2016). Thirty-six percent of children reported consuming RTE cereal. RTE cereal eaters consumed the same number of calories as non-eaters but had higher intakes of total carbohydrates, total sugar, fiber, calcium, iron, magnesium, potassium, zinc, vitamin A, thiamin, riboflavin, niacin, vitamin B_6_, folate, vitamin B_12_, and vitamin D, as well as lower intakes of total fat and saturated fat (*p* ≤ 0.0007). We also found that children who consumed RTE cereal had 29% higher total dairy intake (*p* < 0.0001) and 61% higher whole grain intake (*p* < 0.0001). Lastly, children who ate RTE cereal had higher diet quality than the children that did not eat RTE cereal, as shown by Healthy Eating Index 2015 total score (52.6 versus 47.7, *p* < 0.0001). Therefore, consumption of whole-grain fortified RTE cereals should be encouraged as part of healthy dietary patterns for children.

## 1. Introduction

Ready-to-eat (RTE) cereal is a popular breakfast food for children. Previous results have shown that children who eat RTE cereal, at any time or as a part of breakfast, have higher nutrient intakes, including nutrients of public health concern (calcium, fiber, potassium, and vitamin D) [1], higher whole grain intake, and are more likely to meet nutrition recommendations [2,3,4,5,6,7,8,9]. In addition, inverse associations with health outcomes such as overweight/obesity, hypertension, blood lipids, type 2 diabetes, and all-cause mortality have been observed with RTE cereal intake [10,11]. While there have been controversies related to the nutritional benefits of RTE cereal, the balance of the published science [10], coupled with the recommendations within the 2015 Dietary Guidelines for Americans (DGA) [1], support whole-grain fortified RTE cereal as positively contributing to nutrient intakes and overall diet quality.

However, there are no recent data on the associations between RTE cereal consumption and nutrient intakes in the U.S. Potentially, shifts in RTE cereal intake patterns in either type, amount, or prevalence of consumption could have occurred that could impact this association in children and adolescents. Furthermore, the 2020 DGA will, for the first time, develop nutrition recommendations for infants and toddlers from birth to 24 months [12]. RTE cereal is a popular food among infants and toddlers, yet, to our knowledge, there is limited information available on the role that RTE cereal plays in the diet in this age group [13].

Therefore, we sought to investigate how RTE cereal was associated with nutrient intakes, nutritional adequacy, food group intake, and overall dietary quality among children from 6 months to 17 years of age, comparing those who do not eat cereal to those that reported eating cereal using the latest U.S. nationally representative data, the National Health and Nutrition Examination Survey (NHANES) 2015–2016 [14]. We also report how the prevalence of RTE cereal intake has changed over time using NHANES 2003–2004 through to 2015–2016.

## 2. Materials and Methods 

### 2.1. Data Source and Population

The NHANES 2015–2016 and Food Patterns Equivalents Database (FPED) 2015–2016 data were used for these analyses [14,15]. NHANES 2015–2016 is a nationally representative cross-sectional survey that samples the resident, noninstitutionalized, civilian U.S. population collected by the National Center for Health Statistics of the Centers for Disease Control and Prevention. The NHANES study protocol was approved by the National Center for Health Statistics Ethics Review Board. A parent or guardian provided written informed consent for minors to participate; additionally, children aged 7 to 11 years provided their written informed assent and children aged 12 to 17 years provided their written informed consent to participate. Documentation on methodology and all data are publicly available [16]. NHANES has collected demographic, health, and dietary data on a representative sample of the U.S. population in 2-year cycles using similar methodology since 2003–2004. An in-person household interview collects demographic data through a computer-assisted personal interview. Participants then undergo a health examination in a Mobile Examination Center where a 24-h dietary recall is conducted. A second dietary recall is then conducted through a telephone interview 3 to10 days later. Both the in-person and telephone 24-h recalls use the validated automated multiple-pass method [17,18]. Dietary intake information for children under 6 years of age is gathered from proxy interviews without the child present. For children aged 6 to 8 years the child is present and assists the proxy with the dietary interview. Children aged 9 to 11 years primarily provide the information about their dietary intake themselves but with the help of an assistant/adult familiar with their dietary intake. Lastly, children aged 12 years and over provide information on their dietary intake alone without the assistance of an adult. A proxy can be a parent, grandparent, babysitter, or any other person knowledgeable about the child’s intake [19].

The dietary data was processed using the United States Department of Agriculture (USDA) Food and Nutrient Database for Dietary Studies (FNDDS) to estimate nutrient intake from each reported food and beverage [20]. The FPED 2015–2016 converts the foods and beverages in the FNDDS to the 37 USDA Food Patterns components [15]. 

Participants aged ≥6 months (0.5 years) to <18 years with reliable day-1 dietary recall data were included in this study (*n* = 2969). We categorized participants according to their RTE cereal consumption status (RTE cereal eaters and RTE cereal non-eaters) and according to age group: infants and toddlers (age ≥0.5 years (6 months) to ≤2 years (24 months), *n* = 324); children (>2 years (24 months) to ≤12 years, *n* = 1857); and adolescents (≥13 years to ≤17 years, *n* = 788). While pregnancy and lactation information is collected for participants 12 to 19 years, it is not released because of disclosure risks; therefore, we did not exclude participants based on pregnancy or lactation status.

### 2.2. RTE Cereal Consumption 

Participants were classified into RTE cereal eaters or non-eaters if they reported consuming RTE cereal on day 1 of their 24-h dietary recall. One day of 24-h dietary recall provided an appropriate estimate of population mean intake of nutrients on a given day [21]; allowed for a consistent methodology (in-person dietary interview) to define RTE cereal eaters; and included a larger sample size (588 participants with a valid day-1 dietary recall did not have a reliable, or did not complete, day 2 dietary recall, Appendix A). Additional information on the dietary recall status for the day-1 versus day-2 24-h recalls and a sensitivity analyses using two days of dietary recalls to estimate mean nutrient intake can be found in our Appendix A. We defined RTE cereals using USDA’s What We Eat In America (WWEIA) classification system [20]. WWEIA classifies individual foods into groups according to how they are typically consumed and on their nutrient content. All foods identified by WWEIA as “ready-to-eat cereal, high sugar (>21.2 g/100 g)” and “ready-to-eat cereal, low sugar (≤21.2 g/100 g)” were used to define RTE cereal in the current study which includes RTE cereals with varying levels of whole grain and fortification profiles. Day-1 dietary data from previous cycles of NHANES data (2003–2004 through to 2015–2016) were also examined to calculate the percentages of RTE cereal eaters to describe RTE cereal consumption over time, after the same inclusion and exclusion criteria listed above were applied to each cycle. 

### 2.3. Outcome Variables

#### 2.3.1. Macro- and Micro-nutrients

Total energy intake and nutrient intakes were obtained from day-1 24-h dietary recalls. Nutrients from dietary supplements were not considered in this analysis based on the underlying premise of the 2015–2020 DGA that “nutritional needs should be met primarily from foods” [1]. We selected the nutrients to report based on those that are (a) fortified in RTE cereal and (b) those that were identified as under-consumed nutrients and/or nutrients of public health concern from the 2015–2020 DGA [1].

#### 2.3.2. Percent Contribution of RTE Cereal to Daily Nutrient Intake

We calculated the contribution of RTE cereal to daily nutrient intakes for RTE cereal eaters only and for the total population by taking the percent of the means (i.e., mean nutrient intake from RTE cereal/mean nutrient intake for the day × 100). The contribution of RTE cereal among RTE cereal eaters provides an estimate of the direct contribution that cereal is making to dietary intake separate from differences in diet quality between RTE cereal eaters and non-eaters. The contribution of RTE cereal among the total population provides an estimate of the overall public health impact of RTE cereal consumption.

#### 2.3.3. Food Group Intake

Using the FPED 2015–2016, we reported the mean intake of key food groups for RTE cereal eaters and non-eaters. We used the food groups as reported by FPED, which disaggregates all foods according to standard recipes to calculate the intake of food groups from all foods. Values are reported as cup equivalents or ounce equivalents, which each represent a serving for the respective food groups [15]. We reported intake of the main food groups including total dairy (cup equivalents), total fruit (cup equivalents), total vegetables (cup equivalents), total protein foods (ounce equivalents), and total grain (ounce equivalents). We further reported on key food sub-groups including fluid milk (cup equivalents), intact fruit (cup equivalents), fruit juice (cup equivalents), whole grains (ounce equivalents), and refined grains (ounce equivalents) because these are plausibly associated with RTE cereal consumption. 

#### 2.3.4. Healthy Eating Index 2015 Scores

Dietary quality was measured using the Healthy Eating Index (HEI) 2015 [22], which measures adherence to U.S. dietary guidelines independent of calorie intake (all values are normalized per 1000 kcal). Food group intake was calculated using day-1 dietary data using the FPED 2015–2016 [15]. The maximum HEI score is 100 (highest-quality diet) based on the sum of 13 sub-scores for the intake of total fruits, whole fruit, total vegetables, greens and beans, whole grains, dairy, total protein foods, seafood and plant proteins, fatty acids, refined grains, sodium, added sugar, and saturated fats [22]. 

### 2.4. Covariates 

Age and energy intake were included as continuous variables in the models. Gender (male, female), race/ethnicity (Non-Hispanic White, Non-Hispanic Black, Mexican American, Other Hispanic, and Other), and poverty–income ratio (PIR; ≤1.85, 1.86–3.49, ≥3.50) were included as categorical variables in analysis. The PIR uses a set of income thresholds that vary based on family size and composition to determine who is in poverty. The threshold of 1.85 was chosen because this reflects the threshold for qualifying for reduced prices in the National School Lunch and School Breakfast federal feeding programs [23]. An upper threshold of 3.50 is often used by the National Center for Health Statistics [24]. None of the participants included in the analyses had missing values. 

### 2.5. Statistical Analysis 

SAS 9.3 (SAS Institute, Cary, NC, USA) was used for data analysis. Two-year sample weight and SAS survey procedures were applied. Categorical variables were compared by survey Chi-square tests by RTE cereal consumption status, whereas continuous variables were compared between RTE cereal consumers and non-consumers by survey multivariable linear regression. Data were presented as weighted percentage (according to dietary day-1 study weights published by NHANES 2015–2016 [14]) or least squares means with standard errors. For each hypothesis tested (i.e., differences in nutrient intakes, differences in food groups, and the HEI score), we applied a Bonferroni correction to determine the level of statistical significance. We compared 24 nutrients across 3 age subgroups for a total of 74 comparisons, and therefore we considered a *p*-value ≤ 0.0007 to be statistically significant. For food groups, we compared 10 food groups across 3 age subgroups for 30 comparisons and therefore we considered a *p*-value ≤ 0.002 to be statistically significant. Lastly, we compared 13 sub-scores and the main HEI score across 3 age subgroups for a total of 42 comparisons, and therefore considered a *p*-value ≤ 0.001 to be statistically significant. 

## 3. Results

### 3.1. Prevalence of RTE Cereal Consumption and Demographic Characteristics of RTE Cereal Eaters

Table 1 displays the demographic characteristics of RTE cereal eaters for each age group. RTE cereal eaters were younger compared to RTE cereal non-eaters overall and in each age subgroup, except for infants and toddlers where they were older. The percentage of girls and boys was similar among RTE cereal eaters and non-eaters across all age groups. Self-reported ethnicity did not vary between RTE cereal eaters and non-eaters, except in the infant and toddler group where there was a higher prevalence of Non-Hispanic White RTE cereal eaters and a lower prevalence of Non-Hispanic Black RTE cereal eaters compared to RTE cereal non-eaters. There were no significant differences in PIR bands between RTE cereal eaters and non-eaters. Among children 2 to 12 years, adolescents 13 to 17 years, and all children, RTE cereal eaters were less likely to skip breakfast.

### 3.2. Trends in RTE Cereal Consumption

In 2015–2016, the prevalence of RTE cereal eating was 36% for all children and ranged from 33% for infants and toddlers and 37% for children aged 2 to 12 years. The prevalence of RTE cereal consumption decreased among infants and toddlers from 42% in 2003–2004 to 33% in 2015–2016, and among children decreased from 46% in 2003–2004 to 37% in 2015–2016. Conversely, we observed an increase in the prevalence of RTE cereal consumptions among adolescents from 29% in 2003–2004 to 34% in 2015–2016 (Figure 1 and Appendix A). Overall, among all children 0.5 to 17 years we observed a decrease in the prevalence of RTE cereal consumption from 41% in 2003–2004 to 36% in 2015–2016. Combining all cycles of NHANES together (2003–2016), we found that the prevalence of RTE cereal consumption over this time period was 36% for infants and toddlers, 43% for children, 29% for adolescents, and 38% for all children 0.5 to 17 years (Appendix A).

### 3.3. Differences in Nutrient Intakes for the Total Day between RTE Cereal Eaters and RTE Cereal Non-Eaters

Table 2 presents the results for the percent difference in nutrient intakes between RTE cereal eaters and non-eaters adjusted for demographic characteristics (Model 1) and demographic characteristics plus energy intake (Model 2). Among all children (0.5 to 17 years) RTE cereal eaters consumed the same number of calories as RTE cereal non-eaters (Model 1) and (after adjusting for energy intake, Model 2) had higher intakes of total carbohydrates, total sugar, fiber, calcium, iron, magnesium, potassium, zinc, vitamin A, thiamin, riboflavin, niacin, vitamin B_6_, folate, vitamin B_12_, and vitamin D. Intakes of total fat and saturated fat were significantly lower, and intakes of added sugar, protein, sodium, vitamin C, and vitamin E were not significantly different. We found similar results in a sensitivity analysis using two days of 24-h dietary recalls (Appendix A). These patterns were similar across children and adolescents with two notable exceptions: dietary fiber was not significantly different in children and adolescents and saturated fat was not different in adolescents. Among infants and toddlers, there were fewer differences: only thiamin, vitamin B_6_, and folate were significantly higher in RTE cereal eaters compared to non-eaters, and all other nutrients were not significantly different. As would be expected, intake of total calories and all nutrients appeared to increase across increasing age groups (Appendix A).

We also found that RTE cereal eaters were more likely to meet nutrient intake recommendations, as defined by the estimated average requirement (EAR), than cereal non-eaters (Appendix A). For children ages 2 to 17 years, less than 5% of the population was below the EAR for thiamin, riboflavin, niacin, vitamin B_6_, vitamin B_12_ or iron, regardless of RTE cereal consumption. We saw greater differences in the percent below the EAR between RTE cereal eaters and non-eaters for folate (0% vs. 9%), zinc (2% vs. 11%), vitamin A (3% vs. 35%), vitamin C (13% vs. 25%), calcium (29% vs. 54%), and vitamin D (85% vs. 98%). These patterns were similar by age group, although younger children were overall less likely to be below the EAR compared to adolescents (Appendix A). 

### 3.4. Contribution of RTE Cereal to Daily Nutrient Intakes

The results in Table 2 present the daily diets of RTE cereal eaters compared to non-eaters. Some of the differences observed were due to RTE cereal per se, but also the results of different dietary patterns between RTE cereal eaters and non-eaters. Figure 2 quantifies the contribution of RTE cereal to daily intake both for RTE cereal eaters but also for the total population, which provides an overall public health impact of RTE cereal consumption. Among RTE cereal eaters aged 0.5 to 17 years, RTE cereal contributed 9% of energy intake. It contributed proportionally more to the daily intake of folate, iron, whole grains, vitamin B_6_, vitamin B_12_, thiamin, niacin, vitamin A, zinc, riboflavin, vitamin D, added sugar, fiber, total carbohydrate, and vitamin C. Conversely, RTE cereal contributed proportionally less to the daily intake of saturated fat, total fat, total protein, and potassium. Intakes were within ±2.5 percentage points of the contribution of RTE cereal to energy intake (approximately proportional) for vitamin E, total sugar, magnesium, sodium, and calcium. This pattern was similar for infants and toddlers, children, and adolescents, except that the contribution of RTE cereal to the intakes of infants and toddlers was lower for both energy (6%) and most nutrients. 

When we considered the overall public health impact of RTE cereal consumption by investigating the contribution of RTE cereal to the intakes of the total population, we found that RTE cereal contributed 3% to total energy intakes and proportionally more to intakes of folate, iron, whole grains, vitamin B_6_, vitamin B_12_, vitamin A, thiamin, niacin, zinc, riboflavin, vitamin D, added sugar, and dietary fiber. RTE cereal contributed proportionally (within ±2.5 percentage points of its contribution to energy) to the intake of vitamin C, total carbohydrate, total sugar, vitamin E, magnesium, calcium, sodium, potassium, protein, total fat, and saturated fat. 

Lastly, due to the frequent co-consumption of RTE cereal with milk (86% of RTE cereal is consumed with milk for children 0.5 to 17 years, data not shown), we also calculated the percent contribution to daily intake among RTE cereal eaters for RTE cereal and milk combined (Appendix A). The contribution of RTE cereal to daily intakes of nutrients increased when we also considered the contribution of milk for vitamin B_12_ (from 40% to 53%), vitamin D (from 21% to 47%), vitamin A (from 34% to 45%), riboflavin (from 27% to 40%), calcium (from 8% to 25%), total sugar (from 11% to 18%), total carbohydrate (from 15% to 18%), magnesium (from 10% to 18%), potassium (from 5% to 15%), protein (from 5% to 13%), sodium (from 8% to 11%), saturated fat (from 2% to 10%), and total fat (from 3% to 8%). 

### 3.5. Food Group Intake for RTE Cereal Eaters and RTE Cereal Non-Eaters 

In addition to the differences in nutrient intakes, we also found that children who consumed RTE cereal had higher intakes of recommended food groups, including significantly higher total dairy intake (29% higher, *p* < 0.0001) and higher whole grain intake (61% higher, *p* < 0.0001) (Table 3). This is supported by the frequent co-consumption of RTE cereal with milk as well as the important contribution that RTE cereal makes to whole grain intake in RTE cereal eaters (48% of all whole grain intake). We also found that RTE cereal eaters had significantly lower intake of total protein foods (19% lower, *p* = 0.0001). Looking at subcategories of protein foods, we found that RTE cereal eaters aged 0.5 to 17 years had lower intake of meat, poultry, and seafood (*p* = 0.0005) as well as eggs (*p* = 0.0009), but we found no difference in the intake of nuts and seeds, soybean products, or legumes (Appendix A). 

### 3.6. Healthy Eating Index for RTE Cereal Eaters and RTE Cereal Non-Eaters 

Using the HEI 2015 as a measure of diet quality, we found that children aged 0.5 to 17 years who reported consuming RTE cereal had higher diet quality on that day than children that did not report consuming RTE cereal (52.6 versus 47.7, *p* < 0.0001) (Table 4). When we examined the subscores, we found that children who consumed RTE cereal scored higher on the whole grain, dairy, refined gain, sodium, and saturated fat subscores. This pattern was similar among children aged 2 to 12 years and adolescents aged 13 to 17 years. There was no significant difference between RTE cereal eaters and non-eaters for overall diet quality for infants and toddlers.

## 4. Discussion

RTE cereal was developed in the 1890s as a health food, and due to its popularity the product offerings of the RTE cereal category have expanded in the past 120 years. Still, the core health principles of RTE cereal remain: an often whole-grain and fiber-rich cereal fortified with key nutrients that forms the basis of a breakfast meal, particularly when consumed with milk and fruit [25]. RTE cereal is popular among children and, in part due to this, its nutritional value has been closely scrutinized. Previous research in the U.S. [2,6], Europe [26,27], Canada [28], and elsewhere [10,29,30] has consistently shown positive associations between RTE cereal consumption in children and measures of dietary quality; however, it is possible that these associations may change due to shifts in overall dietary patterns of children, or changes in the prevalence of consumption or nutritional composition of RTE cereal. However, using the most recent nationally representative food intake data in the US, we found that RTE cereal continues to provide key nutrients in the diets of children, and children who consumed RTE cereal had higher overall diet quality compared to children that did not.

### 4.1. Cereal Consumption over Time

The prevalence RTE cereal consumption was high in 2015–2016, with 36% of children 0.5 to 17 years eating RTE cereal on any given day in the U.S. This was similar for all age groups and to reports from Canada [31], and slightly higher than in the UK [32]. When we looked at trends over time in RTE cereal consumption, there has been a decrease in the prevalence of RTE cereal consumption since 2003–2004, where the prevalence was 41%. This decrease was seen among infants and toddlers as well as children aged 2 to 12 years. In adolescents, it appears as though the consumption of RTE cereal may be increasing. In 2003–2004, the prevalence of RTE consumption for adolescents aged 13 to 17 years was 29%, considerably lower compared to younger children. This reached a nadir of 24% in 2011–2012, after which the prevalence of RTE consumption increased to 28% in 2013–2014 and 34% in 2015–2016. To our knowledge, this is the first time that trends in RTE cereal consumption have been published, and it will be important to continue to monitor how RTE cereal consumption changes over time in each age subgroup and to understand the driving forces behind these changes.

### 4.2. Nutrient and Food Group Intakes in Infants and Toddlers

The 2015–2020 DGA highlighted whole-grain fortified RTE cereal as an important source of whole grains and nutrients in the diets of Americans [1]. The 2015–2020 DGA were directed to the general population 2 years of age and older. For the first time, the upcoming 2020–2025 DGA will include recommendations for infants and toddlers from birth to 24 months. RTE cereal is a popular early food among infants and toddlers (we found that 33% of infants and toddlers aged 0.5 to 2 years reported consuming RTE cereal), and therefore information on its contribution to the dietary intakes of this age group are particularly relevant to the upcoming DGA. We found that infants and toddlers aged 0.5 to 2 years who consumed RTE cereal were slightly older than those not consuming RTE cereal, which aligns with the typical age and developmental milestones at which this type of complementary food would be introduced [33]. The U.S.-based 2016 Feeding Infants and Toddlers study reported that 20% of 6 to 11.9 month-olds, 50% of 12 to 17.9 month-olds, and 58% of 18 to 23.9 month-olds were reported as consuming “family cereal”, which includes both RTE and hot cereal [34]. 

Energy intake between RTE cereal eaters and non-eaters was not different for infants and toddlers, nor were most nutrients (except for the B vitamins, which were higher in RTE cereal eaters). RTE cereal contributed to 6% of energy intake in this age group but 40% of iron intake (a key nutrient for this age group), 48% of whole grain intake, and over 30% of the intake of several B-vitamins, vitamin A, and zinc. Iron intake is particularly important for infants and toddlers, as inadequate iron intake at this age may be associated with delayed cognitive development [35]. Iron-fortified baby cereals are recommended as a first complementary food, and as infants and toddlers age and reach appropriate developmental milestones, RTE cereal can become an important source of iron and other nutrients in their diets [13].

The 2020–2025 DGA may recommend a specific dietary pattern for the birth-to-24 months age group, and it would be congruent with their approach in older children if recommendations were made to encourage half of all grain intake to come from whole grains. The Healthy U.S. Eating Pattern at the 1000 calorie level, which the 2015–2020 DGA noted were targeted toward children aged 2 to 8 (along with the 1200 and 1400 calorie levels), recommends that 1.5 oz eq. of whole grains per day are consumed [1]. In our study, we found that infant and toddler RTE cereal eaters ate an average of 0.7 oz eq. of whole grains per day whole and RTE cereal non-eaters consumed 0.4 oz eq., and that RTE cereal contributed to 48% of whole grain intake among RTE cereal eaters. Therefore, infants and toddlers are likely falling short on whole grain intake, and RTE cereal is an important source of whole grains for this age group. 

Added sugar will likely be another important area to be addressed. After adjusting for age, demographic characteristics, and calorie intake, we found no significant difference in added sugar intake between RTE cereal eaters and non-eaters for infants and toddlers aged 0.5 to 2 years. Both RTE cereal eaters and non-eaters consumed 3.9 tsp (or about 16 g) of added sugar per day, which is approximately 6% of energy intake for this age group. Furthermore, RTE cereal contributed 16% of added sugar intake for this age group, which is less than 1 tsp eq./day from RTE cereal. 

### 4.3. Nutrient and Food Group Intakes in Children and Adolescents

Children 2 to 12 years who ate RTE cereal displayed an overall improved dietary pattern compared to children who did not consume RTE cereal. They had higher intake of nutrients of public health concern and under-consumed nutrients (from the 2015–2020 DGA), including 60% higher vitamin D intake, 15% higher calcium intake, 8% higher potassium intake, 52% higher vitamin A, and 71% higher iron. RTE cereal was a key contributor to these nutrients, providing 52% of iron intake, and when consumed with milk, 48% of vitamin D intake, 26% of calcium intake, 15% of potassium, and 45% of vitamin A. This is all while RTE cereal eaters did not consume more saturated fat, sodium, or added sugar. Lastly, dairy consumption—driven mainly by a half serving more per day of fluid milk—and whole grain consumption were higher among children who ate RTE cereal compared to those that did not. RTE cereal is the number one source of whole grains for children aged 2–12 years, providing 46% of their daily whole grain intake. Overall, the nutrient density of cereal and its frequent co-consumption with milk led to a higher overall diet quality for children who consumed RTE cereal compared to those that did not. These results are congruent with the literature on RTE cereal and diet quality [10]. 

We found similar results in adolescents: RTE cereal eaters had higher intake of calcium, vitamin D, iron, and vitamin A, and RTE cereal was a key contributor to the daily intake of these nutrients. Sodium, saturated fat, and added sugar intake were not different between adolescent RTE cereal eaters and non-eaters. Additionally, similar to children, adolescents who ate RTE cereal had over half a serving higher intake per day of dairy and whole grains. At the same time, we also observed that adolescent RTE cereal eaters ate an average of 1.2 fewer servings per day of meat, poultry, and seafood and 0.2 fewer servings per day of eggs. The 2015–2020 DGA noted that meat, poultry, and eggs were a food group that is overconsumed—particularly among teenage boys and men [1]. This eating pattern closely reflects that recommended in the 2015–2020 DGA and accounts for the higher HEI score seen in adolescents who ate RTE cereal. This finding on protein food consumption warrants further study as the area of sustainable diets continues to gain prominence.

The adolescent age group is also unique compared to children in that a higher percentage of the population is below the EAR for some key nutrients. We found large discrepancies (i.e., more than 10 percentage points) between adolescents who ate RTE cereals and those that did not in the percent below the EAR for vitamin A (10% vs. 69% below EAR), calcium (43% vs. 73%), folate (0% vs. 25%), vitamin D (77% vs. 98%), zinc (6% vs. 24%), vitamin C (34% vs. 50%), and iron (0% vs. 11%). Adolescence may be a time of particular vulnerability for poor diet [36], and these results suggest that RTE cereal may be an important strategy to increase key food groups and nutrients to prevent possible nutrient deficiencies and improve diet quality. 

### 4.4. Nutrient Intakes across All Age Groups

Similar to the findings by age group, we found that, compared to non-eaters, all children aged 0.5 to 17 years that ate RTE cereal had higher intake of nutrients of public health concern, including fiber, calcium, potassium, vitamin D, and iron. They also had higher intake of under-consumed nutrients, including magnesium and vitamin A. At the same time, RTE cereal eaters when all age groups were combined had lower saturated fat intake and no significant difference in added sugar or sodium intake. Total carbohydrate and total sugar intake were significantly higher in RTE cereal eaters compared to non-eaters, but total carbohydrate was within the acceptable macronutrient distribution range for children [37]. Conversely, there are no intake recommendations in the U.S. for total sugar, only for added sugar. We did find that fiber intake was higher for RTE cereal eaters compared to non-eaters when all age groups were combined, but the difference did not reach statistical significance for any of the age subgroups. 

### 4.5. Public Health Implication

RTE cereal is in a widely consumed food among children and teens, and has long served as a foundational food for breakfast providing important nutrients of need in children’s diets. However, it is also often the subject of criticism by advocacy organizations due to its sugar content. Despite this, fortified whole-grain breakfast cereal remains a food recommended by the DGA, and our findings using the most recent U.S. dietary data support this recommendation. RTE cereal serves as an important food for children and adolescents due to its ability to deliver under-consumed nutrients and food groups (dairy and whole grain) in a convenient, palatable, and cost-effective food. Food manufacturers have a role to play in developing and reformulating RTE cereals to improve their nutritional profile by increasing the content of whole grain, fiber, and other under-consumed nutrients balanced with decreasing nutrients of concern, primarily sugar, while maintaining consumer acceptance. Considering the nutrient density of a food and the role it plays in the overall diet may be more important in building healthier dietary patterns rather than only considering the contribution a food makes to nutrients to limit. Recent findings from the Global Burden of Disease Study found that the top dietary risk factors were not eating enough whole grains, eating too much salt, and not eating enough fruit [38]. The authors noted that “although sodium, sugar, and fat have been the main focus of diet policy debate in the past two decades, our assessment shows that the leading dietary risk factors for mortality are diets high in sodium, low in whole grains, low in fruit, low in nuts and seeds, low in vegetables and low in omega-3 fatty acids; each accounting for more than 2% of global deaths. This finding suggests that dietary policies focusing on promoting the intake of components of diet for which current intake is less than the optimal level might have a greater effect than policies only targeting sugar and fat” [38]. 

### 4.6. Strengths and Limitations

This study has numerous strengths, including the use of a large, nationally representative data set that used a validated method for collecting dietary intake. The RTE cereal category is one of the few food categories in NHANES that includes specific branded information, so there is very detailed food intake data for this particular food category. These findings also represent the impact of RTE cereal as consumed, including all types and brands of cereal and reflects the popularity of different cereal types. The strength of this approach is that it represents the RTE cereal preferences of consumers. However, further research is warranted to examine the nutritional and health impact of consuming cereals with different nutritional profiles. We also included all ages of children, including 6 months to 24 months, and reported our findings by age group.

Despite these strengths, there are some weaknesses that should be acknowledged—our results are cross-sectional and observational in nature. These findings cannot establish causality but instead are meant to provide an overview of the dietary status of those consuming RTE cereal. Our results represent the mean nutrient intake for RTE cereal eaters and non-eaters on a given day and do not represent the usual dietary intake of the participants or habitual RTE cereal intake. Likely, the improved dietary quality we reported is due to both the contribution of RTE cereal itself and a reflection of an overall healthier dietary pattern, including greater prevalence of breakfast consumption, among RTE cereal eaters. While the 2015–2020 DGA recommend consuming most nutrients from foods, some Americans do use vitamin and mineral supplements. We were unable to consider the contribution of supplements to nutrient intakes, as this data was not available at the time this study was conducted. Lastly, we only examined dietary outcomes in this study, due to the limitations (e.g., confounding and reverse causation) of studying the association between a food and health outcomes in a cross-sectional study; further studies should be conducted to determine the associations between RTE cereal and health outcomes, particularly in children. 

## 5. Conclusions

In summary, dietary guidelines and dietary policies should focus on encouraging consumption of whole-grain fortified RTE cereals, which provide key nutrients and are associated with increased whole grain and dairy consumption and higher dietary quality, as an important part of healthy dietary patterns for children.

## Figures and Tables

**Figure 1 nutrients-11-01989-f001:**
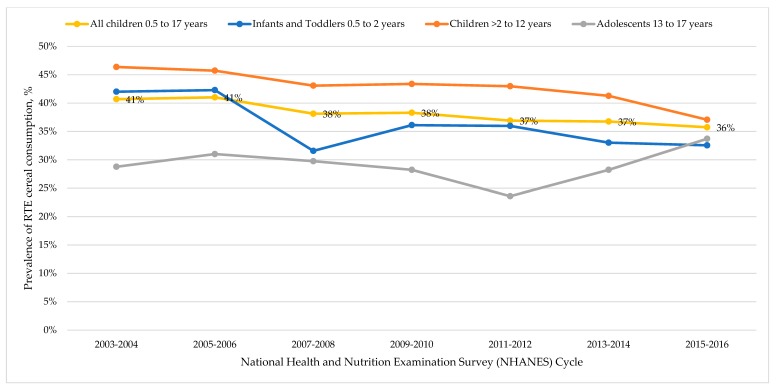
Prevalence of ready-to-eat (RTE) cereal consumption in the National Health and Nutrition Examination Survey (NHANES) from 2003-2004 to 2015-2016 among infants and toddlers 0.5 to 2 years (blue marker and line); children >2 to 12 years (orange marker and line); adolescents aged 13 to 17 years (gray marker and line); and for all children aged 0.5 to 17 years (yellow marker and line). See Appendix A for the number of participants for each age group and time point.

**Figure 2 nutrients-11-01989-f002:**
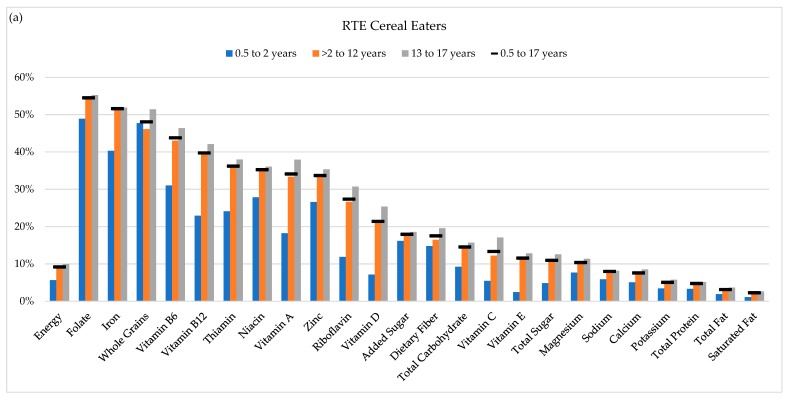
Percent Contribution of RTE cereal to daily intakes: (**a**) for RTE cereal eaters only and (**b**) for the total population.

**Table 1 nutrients-11-01989-t001:** Demographic characteristics of ready-to-eat cereal (RTEC) eaters and non-eaters among children using data from the National Health and Nutrition Examination Survey (NHANES) 2015–2016 ^1^.

		Infants and Toddlers (0.5 to 2 Years)			Children (>2 to 12 Years)			Adolescents (13 to 17 Years)			All Children (0.5 to 17 Years)		
		RTE Cereal Eaters	RTE Cereal Non-Eaters	*p* ^2^	RTE Cereal Eaters	RTE Cereal Non-Eaters	*p* ^2^	RTE Cereal Eaters	RTE Cereal Non-Eaters	*p* ^2^	RTE Cereal Eaters	RTE Cereal Non-Eaters	*p* ^2^
n		88	236		688	1169		242	546		1018	1951	
Mean Age years ± SE		1.4 ± 0.07	1.2 ± 0.03	0.007	6.7 ± 0.2	7.3 ± 0.1	0.007	14.7 ± 0.1	15.2 ± 0.1	0.002	8.7 ± 0.2	9.4 ± 0.2	0.06
Female, n (weighted %)		39 (41%)	113 (42%)	0.89	341 (50%)	593 (49%)	0.65	103 (49%)	279 (51%)	0.71	483 (49%)	985 (49%)	0.81
Race/Ethnicity, n (weighted %)	Mexican American	15 (15%)	45 (15%)	0.007	161 (18%)	232 (14%)	0.36	51 (17%)	121 (16%)	0.85	227 (17%)	398 (15%)	0.56
	Other Hispanic	12 (8%)	30 (9%)		89 (9%)	153 (9%)		33 (10%)	59 (7%)		134 (9%)	242 (8%)	
	Non-Hispanic White	43 (65%)	69 (45%)		192 (48%)	356 (52%)		77 (53%)	157 (54%)		312 (50%)	582 (52%)	
	Non-Hispanic Black	14 (8%)	55 (16%)		148 (14%)	266 (15%)		50 (13%)	123 (14%)		212 (13%)	444 (14%)	
	Other	4 (4%)	37 (15%)		98 (11%)	162 (10%)		31 (8%)	86 (9%)		133 (9%)	285 (10%)	
PIR ≤1.85, n (weighted %)		56 (60%)	152 (61%)	0.76	419 (53%)	662 (44%)	0.10	138 (46%)	332 (47%)	0.99	613 (51%)	1146 (46%)	0.25
PIR >1.85 and ≤3.5, n (weighted %)		14 (18%)	43 (20%)		164 (27%)	268 (27%)		55 (25%)	113 (25%)		233 (26%)	424 (26%)	
PIR >3.5, n (weighted %)		18 (22%)	41 (19%)		105 (21%)	239 (29%)		49 (29%)	101 (28%)		172 (23%)	381 (28%)	
Breakfast skipping ^3^, n (weighted %)		19 (25%)	85 (30%)	0.55	17 (3%)	206 (17%)	<0.0001	20 (8%)	210 (36%)	<0.0001	56 (6%)	501 24%	<0.0001

PIR, poverty-to-income ratio; RTE, ready-to-eat; SE, standard error. ^1^ Data are from the National Health and Nutrition Examination Survey 2015–2016. Children aged 0.5 to 17 years with complete day-1 24-h dietary recalls were included in the analysis. ^2^
*p*-values for continuous variables (i.e., age) are based on *t*-test for surveys; *p* values for categorical variables (% female, race/ethnicity, PIR, and breakfast skipping) are based on chi^2^ for surveys. ^3^ Breakfast was defined as any eating occasion identified as breakfast by the study participants that was ≥50 kcal.

**Table 2 nutrients-11-01989-t002:** Percentage difference between ready-to-eat (RTE) cereal eaters and RTE cereal non-eaters in nutrient intakes, National Health and Nutrition Examination Survey 2015–2016 ^1^.

	Infants and Toddlers (0.5 to 2 Years)				Children (>2 to 12 Years)				Adolescents (13 to 17 Years)				All Children (0.5 to 17 Years)			
	Model 1 ^2^		Model 2 ^3^		Model 1 ^2^		Model 2 ^3^		Model 1 ^2^		Model 2 ^3^		Model 1 ^2^		Model 2 ^3^	
	% diff	*p* ^4^	% diff	*p* ^4^	% diff	*p* ^4^	% diff	*p* ^4^	% diff	*p* ^4^	% diff	*p* ^4^	% diff	*p* ^4^	% diff	*p* ^4^
Energy	−1.8	0.79	NA	NA	–0.2	0.93	NA	NA	−0.1	0.99	NA	NA	0.8	0.74	NA	NA
Total Carbohydrate	0.3	0.97	2.0	0.51	7.9	0.01	8.2	**<0.0001**	10.0	0.04	10.1	**<0.0001**	9.7	0.0008	9.0	**<0.0001**
Total Sugar	−0.1	0.99	1.2	0.78	13.0	0.006	13.3	**0.0004**	12.9	0.06	13.0	0.001	13.1	0.002	12.4	**0.0001**
Added sugar	−3.4	0.79	−0.8	0.94	11.2	0.08	11.6	0.04	3.9	0.59	4.1	0.39	10.7	0.04	9.9	0.02
Fiber	−4.6	0.52	−2.8	0.59	6.9	0.04	7.1	0.005	14.6	0.002	14.7	0.0008	10.4	**0.0004**	9.9	**<0.0001**
Total Fat	−4.7	0.51	−2.7	0.35	−10.1	0.003	−9.9	**<0.0001**	−9.2	0.11	−9.2	**0.0004**	−9.0	0.006	−10.0	**<0.0001**
Saturated Fat	−2.8	0.70	−0.9	0.86	−9.8	0.02	−9.6	**<0.0001**	−3.8	0.54	−3.7	0.31	−7.0	0.05	−8.0	**0.0001**
Protein	-0.3	0.98	1.5	0.82	−0.9	0.76	−0.6	0.69	−8.0	0.16	−8.0	0.02	−1.9	0.54	−2.7	0.15
Calcium	3.9	0.65	5.2	0.50	14.7	0.002	15.0	**<0.0001**	26.3	**0.0002**	26.5	**<0.0001**	18.3	**<0.0001**	17.7	**<0.0001**
Iron	15.9	0.20	16.7	0.14	70.3	**<0.0001**	70.8	**<0.0001**	77.2	**<0.0001**	77.6	**<0.0001**	70.1	**<0.0001**	69.6	**<0.0001**
Magnesium	2.6	0.76	4.3	0.41	6.4	0.08	6.6	0.003	11.0	0.10	11.1	0.02	9.2	0.01	8.5	**0.0003**
Potassium	−0.7	0.93	0.8	0.84	7.6	0.03	7.8	**0.0002**	7.3	0.20	7.4	0.02	8.0	0.02	7.4	**0.0002**
Sodium	−6.1	0.54	−3.6	0.61	−6.4	0.005	−6.2	0.003	−6.9	0.19	−6.9	0.06	−5.4	0.05	−6.2	0.02
Zinc	26.6	0.03	27.6	0.008	42.9	**<0.0001**	43.4	**<0.0001**	38.2	0.002	38.3	**0.0001**	42.4	**<0.0001**	42.1	**<0.0001**
Vitamin A	14.3	0.08	15.0	0.05	51.8	**<0.0001**	52.2	**<0.0001**	68.9	**<0.0001**	69.3	**<0.0001**	53.0	**<0.0001**	52.7	**<0.0001**
Thiamin	20.0	0.03	22.1	**0.0007**	37.9	**<0.0001**	37.1	**<0.0001**	45.6	**<0.0001**	46.6	**<0.0001**	42.1	**<0.0001**	41.6	**<0.0001**
Riboflavin	19.0	0.02	20.1	0.001	36.6	**<0.0001**	37.5	**<0.0001**	49.7	**<0.0001**	49.1	**<0.0001**	40.4	**<0.0001**	40.3	**<0.0001**
Niacin	18.7	0.11	19.9	0.02	34.2	**<0.0001**	34.5	**<0.0001**	28.2	**0.0005**	28.4	**<0.0001**	33.2	**<0.0001**	32.7	**<0.0001**
Vitamin B_6_	35.6	0.008	35.9	**0.0002**	62.7	**<0.0001**	62.7	**<0.0001**	56.6	**<0.0001**	57.6	**<0.0001**	61.1	**<0.0001**	60.8	**<0.0001**
Folate	70.7	**0.0005**	71.2	**<0.0001**	89.1	**<0.0001**	89.8	**<0.0001**	92.1	**<0.0001**	92.6	**<0.0001**	92.5	**<0.0001**	92.5	**<0.0001**
Vitamin B_12_	39.2	0.004	39.7	0.003	79.6	**<0.0001**	79.9	**<0.0001**	83.2	**<0.0001**	83.5	**<0.0001**	81.2	**<0.0001**	80.8	**<0.0001**
Vitamin C	−2.4	0.86	−1.2	0.91	17.4	0.002	17.6	0.002	16.3	0.16	16.4	0.10	15.5	0.01	15.1	0.005
Vitamin D	14.9	0.13	15.7	0.09	59.3	**<0.0001**	59.7	**<0.0001**	113.3	**<0.0001**	113.3	**<0.0001**	63.8	**<0.0001**	63.6	**<0.0001**
Vitamin E	−20.4	0.09	−18.8	0.04	0.6	0.90	0.9	0.80	1.4	0.89	1.6	0.84	−0.6	0.92	−1.4	0.71

% diff., percent difference; NA, not applicable. ^1^ Data are from the National Health and Nutrition Examination Survey 2015–2016. Children aged 0.5 years to 17 years with complete day-1 24-h dietary recalls were included in the analysis. RTE cereal eaters were defined as those that reported consuming any quantity of RTE cereal on their day-1 24-h recall. ^2^ Model 1 was adjusted for gender, ethnicity, poverty-to-income ratio, and age. ^3^ Model 2 was adjusted for the variables in model 1 and total daily energy intake (kcal). ^4^ We applied a Bonferroni correction to set our statistical level of significance based on comparisons of 24 nutrients across 3 age groups: 0.05/24 × 3 = 0.0007; *p-*values that met the level of statistical significance are in bold text.

**Table 3 nutrients-11-01989-t003:** Food group intake for children aged 0.5 to 17 years, by RTE cereal eating status from the Food Patterns Equivalents Database 2015–2016 ^1^.

	Infants and Toddlers (0.5 to 2 Years)			Children (>2 to 12 Years)			Adolescents (13 to 17 Years)			All Children (0.5 to 17 Years)		
	RTEC Eaters ^2^	RTEC Non-Eaters	*p*-Value ^3^	RTEC Eaters ^2^	RTEC Non-Eaters	*p*-Value ^3^	RTEC Eaters ^2^	RTEC Non-Eaters	*p*-Value ^3^	RTEC Eaters ^2^	RTEC Non-Eaters	*p*-Value ^3^
Total dairy (cup eq.)	2.1 ± 0.18	1.7 ± 0.13	0.15	2.2 ± 0.18	1.8 ± 0.14	**<0.0001**	2.3 ± 0.09	1.6 ± 0.08	**<0.0001**	2.2 ± 0.17	1.7 ± 0.15	**<0.0001**
Fluid milk (cup eq.)	1.8 ± 0.10	1.3 ± 0.21	0.05	1.5 ± 0.13	1.0 ± 0.09	**<0.0001**	1.4 ± 0.15	0.7 ± 0.12	**<0.0001**	1.5 ± 0.09	1.0 ± 0.08	**<0.0001**
Total fruit (cup eq.)	1.0 ± 0.10	0.9 ± 0.08	0.76	1.1 ± 0.14	1.0 ± 0.12	0.33	1.1 ± 0.15	0.9 ± 0.12	0.03	1.1 ± 0.07	1.0 ± 0.06	0.01
Whole fruit (cup eq.)	0.6 ± 0.06	0.6 ± 0.06	0.32	0.7 ± 0.13	0.7 ± 0.12	0.36	0.8 ± 0.15	0.5 ± 0.12	0.03	0.7 ± 0.06	0.6 ± 0.06	0.04
Fruit juice (cup eq.)	0.3 ± 0.08	0.3 ± 0.05	0.86	0.4 ± 0.03	0.4 ± 0.03	0.98	0.3 ± 0.07	0.3 ± 0.07	0.83	0.4 ± 0.03	0.3 ± 0.03	0.65
Vegetables (cup eq.)	0.5 ± 0.09	0.6 ± 0.10	0.06	0.9 ± 0.07	0.9 ± 0.07	0.50	0.9 ± 0.13	1.0 ± 0.11	0.03	0.8 ± 0.06	0.9 ± 0.07	0.27
Total protein foods including legumes (oz eq.)	1.3 ± 0.14	1.8 ± 0.10	0.02	3.7 ± 0.15	4.3 ± 0.12	0.009	4.1 ± 0.24	5.7 ± 0.25	**<0.0001**	3.6 ± 0.11	4.4 ± 0.10	**0.0001**
Total Grains (oz eq.)	2.7 ± 0.17	2.6 ± 0.14	0.7	6.2 ± 0.15	6.5 ± 0.12	0.15	7.9 ± 0.23	7.7 ± 0.20	0.50	6.3 ± 0.13	6.4 ± 0.11	0.55
Whole grains (oz eq.)	0.7 ± 0.07	0.4 ± 0.05	0.006	1.1 ± 0.10	0.8 ± 0.09	**<0.0001**	1.4 ± 0.20	0.7 ± 0.13	**0.002**	1.2 ± 0.10	0.7 ± 0.07	**<0.0001**
Refined Grains (oz eq.)	2.0 ± 0.16	2.2 ± 0.12	0.33	5.1 ± 0.16	5.7 ± 0.11	**0.002**	6.5 ± 0.22	7.0 ± 0.17	0.05	5.2 ± 0.12	5.7 ± 0.08	**0.002**

^1^ Data are from the Food Pattern Equivalents Database (FPED) 2015–2016. Children aged 0.5 to 17 years with complete day-1 24-h dietary recalls were included in the analysis. Results were adjusted for demographic characteristics and energy intake. ^2^ RTE cereal eaters were defined as those that reported consuming any quantity of RTE cereal on their day-1 24-h recall. ^3^ We applied a Bonferroni correction to set our statistical level of significance based on comparisons of 10 food groups across 3 age groups: 30 comparisons 0.05/30 = 0.002; *p*-values that met the level of statistical significance are in bold text.

**Table 4 nutrients-11-01989-t004:** Healthy Eating Index score and subscores for children aged 0.5 to 17 years, by RTE cereal eating status from the Food Patterns Equivalents Database 2015–2016 ^1^.

	Maximum Score	Infants and Toddlers (0.5 to 2 Years)			Children (>2 to 12 Years)			Adolescents (13 to 17 Years)				All Children (0.5 to 17 Years)		
		RTEC Eaters ^2^	RTEC Non-Eaters	*p*-Value ^3^	RTEC Eaters ^2^	RTEC Non-Eaters	*p*-Value ^3^		RTEC Eaters ^2^	RTEC Non-Eaters	*p*-Value ^3^	RTEC Eaters ^2^	RTEC Non-Eaters	*p*-Value ^3^
Total vegetables	5	1.9 ± 0.22	2.4 ± 0.16	0.03	2.3 ± 0.10	2.2 ± 0.07	0.62		2.2 ± 0.11	2.4 ± 0.10	0.15	2.2 ± 0.07	2.3 ± 0.05	0.42
Greens and beans	5	0.6 ± 0.36	1.0 ± 0.10	0.32	1.0 ± 0.11	1.1 ± 0.10	0.66		0.9 ± 0.15	1.0 ± 0.11	0.59	1.0 ± 0.09	1.0 ± 0.06	0.39
Total fruit	5	3.7 ± 0.21	3.4 ± 0.17	0.39	2.8 ± 0.14	2.7 ± 0.09	0.77		2.6 ± 0.15	2.1 ± 0.10	0.02	2.8 ± 0.10	2.6 ± 0.07	0.08
Whole fruit	5	3.7 ± 0.25	3.3 ± 0.20	0.32	2.7 ± 0.18	2.7 ± 0.12	0.87		2.5 ± 0.20	1.9 ± 0.11	0.01	2.7 ± 0.11	2.5 ± 0.10	0.09
Whole grains	10	3.9 ± 0.31	2.3 ± 0.25	0.003	4.0 ± 0.16	2.6 ± 0.15	**<0.0001**		4.2 ± 0.36	2.1 ± 0.26	0.002	4.0 ± 0.16	2.5 ± 0.14	**<0.0001**
Dairy	10	7.7 ± 0.30	6.4 ± 0.26	0.002	7.8 ± 0.22	6.3 ± 0.14	**<0.0001**		7.3 ± 0.17	5.4 ± 0.14	**<0.0001**	7.8 ± 0.16	6.1 ± 0.11	**<0.0001**
Total protein foods	5	2.1 ± 0.18	2.4 ± 0.15	0.11	3.4 ± 0.09	3.8 ± 0.08	0.01		3.4 ± 0.08	3.8 ± 0.09	0.007	3.4 ± 0.06	3.7 ± 0.07	0.008
Seafood and plant proteins	5	0.6 ± 0.18	0.9 ± 0.12	0.25	1.7 ± 0.09	1.7 ± 0.12	0.80		1.8 ± 0.15	1.7 ± 0.14	0.69	1.7 ± 0.09	1.6 ± 0.10	0.66
Fatty acids	10	1.2 ± 0.37	1.4 ± 0.18	0.66	3.8 ± 0.20	4.0 ± 0.12	0.33		3.6 ± 0.22	4.7 ± 0.19	0.002	3.5 ± 0.14	4.0 ± 0.07	0.02
Sodium	10	8.4 ± 0.52	7.8 ± 0.27	0.35	5.4 ± 0.18	4.7 ± 0.14	**0.001**		4.8 ± 0.34	4.0 ± 0.18	0.04	5.5 ± 0.16	4.8 ± 0.14	0.005
Refined grains	10	8.5 ± 0.35	7.8 ± 0.25	0.12	5.7 ± 0.20	4.7 ± 0.15	**0.0005**		5.2 ± 0.36	4.5 ± 0.23	0.09	5.7 ± 0.15	4.9 ± 0.11	**0.0002**
Saturated fat	10	2.6 ± 0.53	2.6 ± 0.36	0.96	5.8 ± 0.17	4.7 ± 0.19	**0.001**		5.9 ± 0.34	5.4 ± 0.20	0.22	5.6 ± 0.15	4.7 ± 0.10	**0.0004**
Added sugar	10	9.4 ± 0.14	9.2 ± 0.15	0.24	6.5 ± 0.18	7.1 ± 0.13	0.02		6.1 ± 0.34	6.3 ± 0.22	0.60	6.6 ± 0.15	7.1 ± 0.11	0.009
Total HEI 2015 Score	100	54.2 ± 1.25	50.9 ± 0.94	0.06	52.9 ± 0.50	48.3 ± 0.73	**<0.0001**		50.5 ± 0.79	45.3 ± 0.70	**<0.0001**	52.6 ± 0.29	47.7 ± 0.57	**<0.0001**

^1^ Data are from the Food Pattern Equivalents Database (FPED) 2015–2016. Children aged 0.5 years to 17 years with complete day-1 24-h dietary recalls were included in the analysis. Results were adjusted for demographic characteristics and energy intake. ^2^ RTE cereal eaters were defined as those that reported consuming any quantity of RTE cereal on their day-1 24-h recall. ^3^ We applied a Bonferroni correction to set our statistical level of significance based on comparisons of 13 subscores across 3 age groups: 39 comparisons (13 × 3) 0.05/39 = 0.001); *p*-values that met the level of statistical significance are in bold text.

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
