# Peer review of "Association between Ready-to-Eat Cereal Consumption and Nutrient Intake, Nutritional Adequacy, and Diet Quality among Infants, Toddlers, and Children in the National Health and Nutrition Examination Survey 2015–2016"

_nutrients, 2019, doi:10.3390/nu11091989_

Round 1

Reviewer 1 Report

The present manuscript provides a descriptive analysis of ready to eat cereal, nutrient and overall intakes among children and adolescents in the US, at present and over time. The authors conclude that cereal represents an important source of whole grains and various micronutrients in the diet of young Americans. 

This is a straightforward study and the manuscript is well written, if lengthy. The statistical analyses appear to have been conducted adequately and the conclusions are supported by the results presented. It would have been interesting to see whether cereal non-eaters are also breakfast skippers, as ready to eat cereal may simply be a proxy for breakfast and, therefore, the generally positive associations noted by the authors may simply be due to children consuming breakfast. 

There are a few typos and missing words throughout the manuscript, some of which I have highlighted below. Otherwise, the manuscript is suitable for publication in its present form. 

Line 70. Gather --> gathered

Line 120. Intake --> intact?

Line 164. Was --> were.

Line 174. After 38%, a word is missing. 

Line 207. Adolescents is misspelled. 

Line 292. "This the first, to our knowledge." The word "time" should be added after "first."

Line 365. "cereal in" --> "cereal is"

Author Response

Comments and Suggestions for Authors

The present manuscript provides a descriptive analysis of ready to eat cereal, nutrient and overall intakes among children and adolescents in the US, at present and over time. The authors conclude that cereal represents an important source of whole grains and various micronutrients in the diet of young Americans. 

This is a straightforward study and the manuscript is well written, if lengthy. The statistical analyses appear to have been conducted adequately and the conclusions are supported by the results presented. It would have been interesting to see whether cereal non-eaters are also breakfast skippers, as ready to eat cereal may simply be a proxy for breakfast and, therefore, the generally positive associations noted by the authors may simply be due to children consuming breakfast. 

We appreciate your suggestion to also report the percent of cereal eaters that eat/skip breakfast. We have added the prevalence of breakfast skipping to table 1 (line 184). As suggested, the prevalence of breakfast skipping was significantly higher among RTE cereal non-eaters. We added a note to the discussion, at line 433 to 434 in the “Strengths and Limitations” section, that improved dietary quality we reported in our study may be “due to both the contribution of RTE cereal itself and a reflection of an overall healthier dietary pattern, including greater prevalence of breakfast consumption, among RTE cereal eaters.”

There are a few typos and missing words throughout the manuscript, some of which I have highlighted below. Otherwise, the manuscript is suitable for publication in its present form. 

We appreciate the thorough review and have corrected all of the errors noted below.

Line 70. Gather --> gathered  

Line 120. Intake --> intact?

Line 164. Was --> were.

Line 174. After 38%, a word is missing. 

Line 207. Adolescents is misspelled. 

Line 292. "This the first, to our knowledge." The word "time" should be added after "first."

Line 365. "cereal in" --> "cereal is"

Reviewer 2 Report

Overall I feel this paper is an interesting read and important contribution to the literature.  

Please see below for specific comments..

L 70 - term 'gather' should be gathered

L 81 - an explanation of why only day-1 data was used rather than both would be beneficial. Best not to simply defer to other papers but justify within the context of this study.

L 90 - I think you should actually include the definition here rather than simply refer to another paper, given that it is the focus of the study; especially important for the non-American reader who may not be familiar with the definition of RTEC.

L91 - Data from the NHANES cycle 2003-2004 to 2015-2016 were compared.  This is a whole other objective and should be presented in the introduction as a secondary (or simply additional) objective of the research.

L 99 - the point re excluding supplements.  I agree with the position that nutrition is best supported through food but nutrients are also consumed via supplements and contribute to total nutrient intake.  I feel this omission is problematic.  I would have preferred to see a sensitivity analysis of the data with supplements included and discussion around this, rather than simply excluding this data.  At the very least a comment is needed in the discussion and/or limitations section.

L 133 - The acronym PIR should also be defined here as PIR is used later.

L 135 - A reference is needed regarding the poverty-income ratio and the thresholds. 

L 174 - '38%' - doesn't say what this statistic relates to

L 177 - Percentages for RTEC eaters and non-eaters, infants and toddlers, at beginning of first column do not add up to 100.  This table is also difficult to read as numbers and percentages flow over two lines and hence don't line up neatly on each row.  I think this is an editorial issue.

L 203 - assuming that the <5% relates to all of the nutrients?  or could state that less than 5% were below the EAR for....x,y,z...

L 207 - spelling error (adolescents)

L 224 - perhaps state '2.5 percentage points'...rather than 2.5 points (or simply be consistent with how you use this term throughout the article)

L 270 - need a reference for the statement that RTEC was developed in 1890s as a healthy food.

L 272 - RTEC is not always whole grain.

Section 4.1 - as mentioned earlier, this should be stated as an objective in the introduction

L 327 - also worth commenting that there were no differences in added sugar intake between groups

L 339 - also noteworthy is that total sugar and carbohydrates were higher in RTEC consumers

L - 352 to 355- I think a more balanced argument regarding protein is needed here.  The assumption is that this outcome is a positive finding but not necessarily, it would depend on the reasons for the lesser protein intake, which key food sources of protein differ and what this all means for overall diet quality and health.  Also, you are linking this result to just meat, poultry and seafood but are they the only foods included in this category?  Does the category also include plant based protein?  (Line 112, section 2.3.3 - food group intake).. It might help to include a bit more detail in this earlier section regarding foods included in each food category; it would enrich the readers understanding of this result and potentially other findings.

L 364 - I feel that a more robust discussion is warranted on this section on the public health implications.  The current section provides a very one-sided argument with no real attempt to acknowledge alternative perspectives regarding the broader debate around cereals, nor synthesize perspectives.  Acknowledgement of the role of the food industry in reformulating products to improve their nutritional quality is warranted.

Also, more broadly, the discussion does not address that total sugar was higher in children and 'all children' and that there were no differences in fibre between RTEC eaters and non-eaters across all age groups.  This latter finding is particularly interesting as I would have expected to see higher fibre intakes in RTEC consumers.  Comment on this is warranted.

The discussion also fails to acknowledge that the quality of RTEC ranges substantially across the category, from highly refined-sugar laden, to whole grain, nutrient-dense choices. This study does not account for this variation (limitation).  Further research is warranted to examine the nutritional and health impact of consuming these different types of cereals.

Further, while I agree that overall the results show the important contribution of cereals to nutrient intakes, these results are only theoretical as actual total nutrient intake may look very different if supplements were factored into the analysis.  I don't think this should be ignored.

L 400 - I agree that dietary policies and guidelines should encourage whole grain-RTEC consumption but the main findings of this report pertain to RTEC across both refined and whole grain, and fortified and non-fortified categories, don't they? (a definition at Line 90 would help in answering this questions)

There is no label on figure 1.

Author Response

Comments and Suggestions for Authors

Overall I feel this paper is an interesting read and important contribution to the literature.  

Please see below for specific comments..

L 70 - term 'gather' should be gathered This has been corrected.

L 81 - an explanation of why only day-1 data was used rather than both would be beneficial. Best not to simply defer to other papers but justify within the context of this study.

We included the following clarification at lines 92 to 94: “We used only day 1 24-h recalls as this allowed for a consistent methodology (in-person dietary interview) to define RTE cereal eaters and included a larger sample size (some participants did not complete day 2 dietary recalls).”

L 90 - I think you should actually include the definition here rather than simply refer to another paper, given that it is the focus of the study; especially important for the non-American reader who may not be familiar with the definition of RTEC.

We have included more details at lines 95-99 on how ready-to-eat cereal was defined in the current study. We relied entirely on the food classification methodology developed by the United States Department of Agriculture (USDA) and used by NHANES:

“WWEIA classifies individual foods into groups according to how they are typically consumed and on their nutrient content. All foods identified by WWEIA as “ready-to-eat cereal, high sugar (>21.2g/100g)” and “ready-to-eat, low sugar (£21.2g/100g)” were used to define RTE cereal in the current study which includes RTE cereals with varying levels of whole grain and fortification profiles.”

L91 - Data from the NHANES cycle 2003-2004 to 2015-2016 were compared.  This is a whole other objective and should be presented in the introduction as a secondary (or simply additional) objective of the research.

Thank you for this suggestion. We have added this additional objective to our introduction at lines 56-57.

L 99 - the point re excluding supplements.  I agree with the position that nutrition is best supported through food but nutrients are also consumed via supplements and contribute to total nutrient intake.  I feel this omission is problematic.  I would have preferred to see a sensitivity analysis of the data with supplements included and discussion around this, rather than simply excluding this data.  At the very least a comment is needed in the discussion and/or limitations section.

We sought to use the most recent dietary data for this study which is from NHANES 2015-2016. However, information on contribution of nutrients from supplements is not currently available from NHANES 2015-2016; therefore, we were unable to conduct a sensitivity analysis including nutrients from supplements. Per your suggestion, we have added this as a limitation of the study at lines 434 to 437.

L 133 - The acronym PIR should also be defined here as PIR is used later.

This has been added at line 141

L 135 - A reference is needed regarding the poverty-income ratio and the thresholds. 

Two references have been added at lines 145 and 146.

L 174 - '38%' - doesn't say what this statistic relates to

This has been corrected at lines 183-184.

L 177 - Percentages for RTEC eaters and non-eaters, infants and toddlers, at beginning of first column do not add up to 100.  This table is also difficult to read as numbers and percentages flow over two lines and hence don't line up neatly on each row.  I think this is an editorial issue.

We have removed the percentage of RTE cereal eaters and non-eaters to improve the readability of the table as all other percentages add to 100 by adding the rows rather than adding across the columns. Due to space constraints and the volume of data in the table, it was not possible to fit both the number and weighted percentage on the same line. Per your suggestion, perhaps this can be addressed during the final formatting of the manuscript.

L 203 - assuming that the <5% relates to all of the nutrients?  or could state that less than 5% were below the EAR for....x,y,z...

This was referring to less than 5% of the population being below the EAR for the nutrients listed. We have modified the sentence to at lines 213-215.

L 207 - spelling error (adolescents) Corrected

L 224 - perhaps state '2.5 percentage points'...rather than 2.5 points (or simply be consistent with how you use this term throughout the article)

We changed the term to “percentage points” at line 236

L 270 - need a reference for the statement that RTEC was developed in 1890s as a healthy food.

The entire first 4 lines of the discussion are supported by reference 23: Severson, K. A Short History of Cereal. The New York Times. Available online: https://www.nytimes.com/interactive/2016/02/22/dining/history-of-cereal.html) (accessed on June 25 2019).

L 272 - RTEC is not always whole grain.

While the majority of RTE cereals are whole grain rich, you are correct that not all cereals are and there are numerous popular brands that are not whole grain rich. Therefore, we have changed the statement to read that the core health principles of RTE cereal are “an often whole grain and fiber-rich cereal fortified with key nutrients that forms the basis of a breakfast meal…” at line 291.

Section 4.1 - as mentioned earlier, this should be stated as an objective in the introduction

Thank you. This has been added at lines 56 and 57 of the introduction.

L 327 - also worth commenting that there were no differences in added sugar intake between groups

Thank you. We agree that this is important information to add in this context and have included it at lines 348-350.

L 339 - also noteworthy is that total sugar and carbohydrates were higher in RTEC consumers

We have included a discussion on total sugar and carbohydrates at lines 387 to 398 when discussing results in all age groups pooled:

Similar to the findings by age group, we found that all children 0.5 to 17 y that ate RTE cereal, compared to non-eaters, had higher intake of nutrients of public health concern including fiber, calcium, potassium, vitamin D, and iron. They also had higher intake of under-consumed nutrients including magnesium and vitamin A. At the same time, RTE cereal eaters when all age groups were combined had lower saturated fat intake and no significant difference in added sugar or sodium intake. Total carbohydrate and total sugar intake were significantly higher in RTE cereal eaters compared to non-eaters but total carbohydrate was within the Acceptable Macronutrient Distribution Range for children [35]. Conversely, there are no intake recommendations in the U.S. for total sugar, only for added sugar. We did find that fiber intake was higher for RTE cereal eaters compared to non-eaters when all age groups were combined but the difference did not reach statistical significance for any of the age subgroups.

L - 352 to 355- I think a more balanced argument regarding protein is needed here.  The assumption is that this outcome is a positive finding but not necessarily, it would depend on the reasons for the lesser protein intake, which key food sources of protein differ and what this all means for overall diet quality and health.  Also, you are linking this result to just meat, poultry and seafood but are they the only foods included in this category?  Does the category also include plant-based protein?  (Line 112, section 2.3.3 - food group intake).. It might help to include a bit more detail in this earlier section regarding foods included in each food category; it would enrich the readers understanding of this result and potentially other findings.

The total protein category includes both animal sources of protein (meat, poultry, seafood, eggs) and plant-based proteins (legumes and nuts and seeds). We agree that this is an interesting finding worth further discussion. We have now included additional information on the subcategories of protein foods as supplemental table 5 and reference these findings in the results at lines 267-269. RTE cereal eaters consume less protein foods overall. Among the subcategories of protein foods, meat, poultry and seafood and egg intake was lower in RTE cereal eaters. There were no differences in other sources of protein including eggs, nuts and seeds, soybean products, and legumes, although intake of these food groups in generally low. Our statement in the discussion (lines 374-375) is also supported by the 2015-2020 DGAs which evaluated the intake of protein food subcategories and concluded that meat, poultry, and eggs were overconsumed by adolescent boys and men.

L 364 - I feel that a more robust discussion is warranted on this section on the public health implications.  The current section provides a very one-sided argument with no real attempt to acknowledge alternative perspectives regarding the broader debate around cereals, nor synthesize perspectives.  Acknowledgement of the role of the food industry in reformulating products to improve their nutritional quality is warranted.

We agree and have added the following statement at lines 406 to 410 of the discussion:

“Food manufacturers have a role to play in developing and reformulating RTE cereals to improve their nutritional profile by increasing the content of whole grain, fiber, and other under-consumed nutrients balanced with decreasing nutrients of concern, primarily sugar, while maintaining consumer acceptance.”

Also, more broadly, the discussion does not address that total sugar was higher in children and 'all children' and that there were no differences in fibre between RTEC eaters and non-eaters across all age groups.  This latter finding is particularly interesting as I would have expected to see higher fibre intakes in RTEC consumers.  Comment on this is warranted.

We have addressed these findings in the discussion at lines 387 to 398:

4.4 Nutrient Intakes Across All Age Groups

            Similar to the findings by age group, we found that all children 0.5 to 17 y that ate RTE cereal, compared to non-eaters, had higher intake of nutrients of public health concern including fiber, calcium, potassium, vitamin D, and iron. They also had higher intake of under-consumed nutrients including magnesium and vitamin A. At the same time, RTE cereal eaters when all age groups were combined had lower saturated fat intake and no significant difference in added sugar or sodium intake. Total carbohydrate and total sugar intake were significantly higher in RTE cereal eaters compared to non-eaters but total carbohydrate was within the Acceptable Macronutrient Distribution Range for children [35]. Conversely, there are no intake recommendations in the U.S. for total sugar, only for added sugar. We did find that fiber intake was higher for RTE cereal eaters compared to non-eaters when all age groups were combined but the difference did not reach statistical significance for any of the age subgroups.

The discussion also fails to acknowledge that the quality of RTEC ranges substantially across the category, from highly refined-sugar laden, to whole grain, nutrient-dense choices. This study does not account for this variation (limitation).  Further research is warranted to examine the nutritional and health impact of consuming these different types of cereals.

Thank you for this important contribution. We have added this consideration to the Strengths and Limitations section at lines 427-429.

Further, while I agree that overall the results show the important contribution of cereals to nutrient intakes, these results are only theoretical as actual total nutrient intake may look very different if supplements were factored into the analysis.  I don't think this should be ignored.

While supplements have not been considered in this paper, in part due to the lack of data on supplement use and their contribution to nutrients intakes in the NHANES 2015-2016 data, we did examine prevalence of supplement use for both the total population and for RTE cereal eaters alone for a separate publication that is currently under review that used NHANES 2013-2014 data. As shown below, RTE cereal eaters, in general have a similar or higher percentage of supplement use compared to the total population. This gap would be even greater if we compared RTE cereal eaters to non-eaters as the total population included the RTE cereal eaters. Therefore, the results in the current manuscript are likely a conservative representation of the difference in nutrient consumption between RTE cereal eaters and non-eaters. However, we agree that future research should consider supplement use in the analysis and we have added this to the strengths and limitations section at lines 436-439.

Total Population (i.e. RTE cereal eaters and non-eaters)

RTE cereal eaters only

Toddlers 1 to 3 y

Children 4 to 12 y

Adolescents 13 to 18 y

Toddlers 1 to 3 y

Children 4 to 12 y

Adolescents 13 to 18 y

% Day 1 Vitamin or Mineral Supplement Use

30%

24%

18%

34%

25%

26%

Data are from NHANES 2013-2014.

L 400 - I agree that dietary policies and guidelines should encourage whole grain-RTEC consumption but the main findings of this report pertain to RTEC across both refined and whole grain, and fortified and non-fortified categories, don't they? (a definition at Line 90 would help in answering this questions)

Yes, this includes refined grain as well as whole grain rich RTE cereal and includes unfortified and cereals. We are using a well-established, externally developed food classification method, “What We Eat in America”, which is the standard food classification method for NHANES data. We have added additional clarification at lines 95 to 99.

There is no label on figure 1.

There is a label, title and figure legend for figure 1 at lines 193-195.